# TorchMD-NET: Equivariant Transformers for Neural Network based Molecular Potentials

**Philipp Thölke**
Computational Science Laboratory, Pompeu Fabra University,
PRBB, C/ Doctor Aiguader 88, 08003 Barcelona, Spain and
Institute of Cognitive Science, Osnabrück University,
Neuer Graben 29 / Schloss, 49074 Osnabrück, Germany
philipp.thoelke@posteo.de

**Gianni De Fabritiis**
Computational Science Laboratory, Pompeu Fabra University,
C/ Doctor Aiguader 88, 08003 Barcelona, Spain and
ICREA, Passeig Lluis Companys 23, 08010 Barcelona, Spain and
Acellera Labs, C/ Doctor Trueta 183, 08005 Barcelona, Spain
gianni.defabritiis@upf.edu

## Abstract

The prediction of quantum mechanical properties is historically plagued by a trade-off between accuracy and speed. Machine learning potentials have previously shown great success in this domain, reaching increasingly better accuracy while maintaining computational efficiency comparable with classical force fields. In this work we propose TorchMD-NET, a novel equivariant Transformer (ET) architecture, outperforming state-of-the-art on MD17, ANI-1, and many QM9 targets in both accuracy and computational efficiency. Through an extensive attention weight analysis, we gain valuable insights into the black box predictor and show differences in the learned representation of conformers versus conformations sampled from molecular dynamics or normal modes. Furthermore, we highlight the importance of datasets including off-equilibrium conformations for the evaluation of molecular potentials.

## 1 Introduction

Quantum mechanics are essential for the computational analysis and design of molecules and materials. However, the complete solution of the Schrödinger equation is analytically and computationally not practical, which initiated the study of approximations in the past decades (Szabo & Ostlund, 1996). A common quantum mechanics approximation method is to model atomic systems according to density functional theory (DFT), which can provide energy estimates with sufficiently high accuracy for different application cases in biology, physics, chemistry, and materials science. Even more accurate techniques like coupled-cluster exist but both still lack the computational efficiency to be applied on a larger scale, although recent advances are promising in the case of quantum Monte Carlo (Pfau et al., 2020; Hermann et al., 2020). Other methods include force-field and semi-empirical quantum mechanical theories, which provide very efficient estimates but lack accuracy.

The field of machine learning molecular potentials is relatively novel. The first important contributions are rooted in the Behler-Parrinello (BP) representation (Behler & Parrinello, 2007) and the seminal work from Rupp et al. (2012). One of the best transferable machine learning potentials for biomolecules, called ANI (Smith et al., 2017a), is based on BP. A second class of methods, mainly developed in the field of materials science and quantum chemistry, uses more modern graph convolutions (Schütt et al., 2018; Unke & Meuwly, 2019; Qiao et al., 2020; Schütt et al., 2021). SchNet (Schütt et al., 2017b; 2018), for example, uses continuous filter convolutions in a graph network architecture to predict the energy of a system and computes forces by direct differentiation of the neural network against atomic coordinates. Outside of its original use case, this approach has been

extended to coupled-cluster solvers (Hermann et al., 2020) and protein folding using coarse-grained systems (Wang et al., 2019; Husic et al., 2020; Doerr et al., 2021). Recently, other work has shown that a shift towards rotationally equivariant networks (Anderson et al., 2019; Fuchs et al., 2020; Schütt et al., 2021), particularly useful when the predicted quantities are vectors and tensors, can also improve the accuracy on scalars (e.g. energy).

Next to the parametric group of neural network based methods, a nonparametric class of approaches exists. These are usually based on kernel methods, particularly used in materials science. In this work, we will focus on parametric neural network potentials (NNPs) because they have a scaling advantage to large amounts of data, while kernel methods usually work best in a scarce data regime.

Previous deep learning based work in the domain of quantum chemistry focused largely on graph neural network architectures (GNNs) with different levels of handcrafted and learned features (Schütt et al., 2017b; Qiao et al., 2020; Klicpera et al., 2020b; Unke & Meuwly, 2019; Liu et al., 2020; Schütt et al., 2021). For example, Qiao et al. (2020) first perform a low-cost mean-field electronic structure calculation, from which different quantities are used as input to their neural network. Recently proposed neural network architectures in this context usually include some form of attention (Luong et al., 2015) inside the GNN's message passing step (Qiao et al., 2020; Unke & Meuwly, 2019; Liu et al., 2020).

In this work, we introduce TorchMD-NET, an equivariant Transformer (ET) architecture for the prediction of quantum mechanical properties. By building on top of the Transformer (Vaswani et al., 2017) architecture, we are centering the design around the attention mechanism, achieving state-of-the-art accuracy on multiple benchmarks while relying solely on a learned featurization of atomic types and coordinates. Furthermore, we gain insights into the black box prediction of neural networks by analyzing the Transformer's attention weights and comparing latent representations between different types of data such as energy-minimized (QM9 (Ramakrishnan et al., 2014)), molecular dynamics (MD17 (Chmiela et al., 2017) and normal mode sampled data (ANI-1 (Smith et al., 2017b)).

## 2 METHODS

The traditional Transformer architecture as proposed by Vaswani et al. (2017) operates on a sequence of tokens. In the context of chemistry, however, the natural data structure for the representation of molecules is a graph. To work on graphs, one can interpret self-attention as constructing a fully connected graph over input tokens and computing interactions between nodes. We leverage this concept and extend it to include information stored in the graph's edges, corresponding to interatomic distances in the context of molecular data. This requires the use of a modified attention mechanism, which we introduce in the following sections, along with the overall architecture of our equivariant Transformer.

The equivariant Transformer is made up of three main blocks. An embedding layer encodes atom types $Z$ and the atomic neighborhood of each atom into a dense feature vector $x_i$. Then, a series of update layers compute interactions between pairs of atoms through a modified multi-head attention mechanism, with which the latent atomic representations are updated. Finally, a layer normalization (Ba et al., 2016) followed by an output network computes scalar atomwise predictions using gated equivariant blocks (Weiler et al., 2018; Schütt et al., 2021), which get aggregated into a single molecular prediction. This can be matched with a scalar target variable or differentiated against atomic coordinates, providing force predictions. An illustration of the architecture is given in Figure 1.

### 2.1 NOTATION

To differentiate between the concepts of scalar and vector features, this work follows a certain notation. Scalar features are written as $x \in \mathbb{R}^F$, while we refer to vector features as $\vec{v} \in \mathbb{R}^{3 \times F}$. The vector norm $\|\cdot\|$ and scalar product $\langle \cdot, \cdot \rangle$ of vector features are applied to the spatial dimension, while all other operations act on the feature dimension. Upper case letters denote matrices $A \in \mathbb{R}^{N \times M}$.

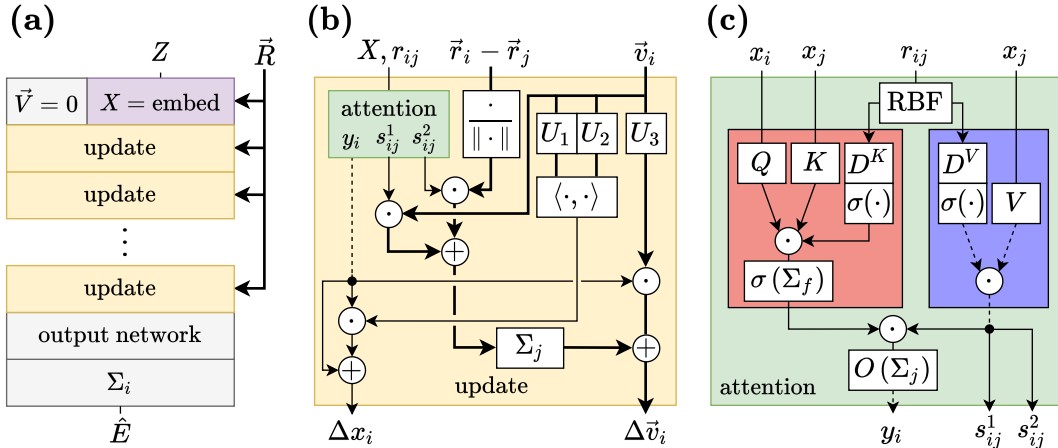

Figure 1: Overview of the equivariant Transformer architecture. Thin lines: scalar features in $\mathbb{R}^F$, thick lines: vector features in $\mathbb{R}^{3 \times F}$, dashed lines: multiple feature vectors. **(a)** Transformer consisting of an embedding layer, update layers and an output network. **(b)** Residual update layer including attention based interatomic interactions and information exchange between scalar and vector features. **(c)** Modified dot-product attention mechanism, scaling values (blue) by the attention weights (red).

## 2.2 EMBEDDING LAYER

The embedding layer assigns two learned vectors to each atom type $z_i$. One is used to encode information specific to an atom, the other takes the role of a neighborhood embedding. The neighborhood embedding, which is an embedding of the types of neighboring atoms, is multiplied by a distance filter. This operation resembles a continuous-filter convolution (Schütt et al., 2017b) but, as it is used in the first layer, allows the model to store atomic information in two separate weight matrices. These can be thought of as containing information that is intrinsic to an atom versus information about the interaction of two atoms.

The distance filter is generated from expanded interatomic distances using a linear transformation $W^F$. First, the distance $d_{ij}$ between two atoms $i$ and $j$ is expanded via a set of exponential normal radial basis functions $e^{\text{RBF}}$, defined as

$$e_k^{\text{RBF}}(d_{ij}) = \phi(d_{ij}) \exp(-\beta_k(\exp(-d_{ij}) - \mu_k)^2) \tag{1}$$

where $\beta_k$ and $\mu_k$ are fixed parameters specifying the center and width of radial basis function $k$. The $\mu$ vector is initialized with values equally spaced between $\exp(-d_{\text{cut}})$ and 1, $\beta$ is initialized as $(2K^{-1}(1 - \exp(-d_{\text{cut}})))^{-2}$ for all $k$ as proposed by Unke & Meuwly (2019). The cutoff distance $d_{\text{cut}}$ was set to 5Å. The cosine cutoff $\phi(d_{ij})$ is used to ensure a smooth transition to 0 as $d_{ij}$ approaches $d_{\text{cut}}$ in order to avoid jumps in the regression landscape. It is given by

$$\phi(d_{ij}) = \begin{cases} \frac{1}{2}\left(\cos\left(\frac{\pi d_{ij}}{d_{\text{cut}}}\right) + 1\right), & \text{if } d_{ij} \leq d_{\text{cut}} \\ 0, & \text{if } d_{ij} > d_{\text{cut}}. \end{cases} \tag{2}$$

The neighborhood embedding $n_i$ for atom $i$ is then defined as

$$n_i = \sum_{j=1}^{N} \text{embed}^{\text{nbh}}(z_j) \odot W^F e^{\text{RBF}}(d_{ij}) \tag{3}$$

with $\text{embed}^{\text{nbh}}$ being the neighborhood embedding function and $N$ the number of atoms in the graph. The final atomic embedding $x_i$ is calculated as a linear projection of the concatenated intrinsic embedding and neighborhood embedding $[\text{embed}^{\text{int}}(z_i), n_i]$, resulting in

$$x_i = W^C \left[\text{embed}^{\text{int}}(z_i), n_i\right] + b^C \tag{4}$$

with $\text{embed}^{\text{int}}$ being the intrinsic embedding function. The vector features $\vec{v}_i$ are initially set to 0.

## 2.3 MODIFIED ATTENTION MECHANISM

We use a modified multi-head attention mechanism (Figure 1c), extending dot-product attention, in order to include edge data into the calculation of attention weights. First, the feature vectors are passed through a layer normalization. Then, edge data, i.e. interatomic distances $r_{ij}$, are projected into two multidimensional filters $D^K$ and $D^V$, according to

$$
\begin{aligned}
D^K &= \sigma(W^{D^K} e^{\text{RBF}}(r_{ij}) + b^{D^K}) \\
D^V &= \sigma(W^{D^V} e^{\text{RBF}}(r_{ij}) + b^{D^V})
\end{aligned}
\tag{5}
$$

The attention weights are computed via an extended dot product, i.e. an elementwise multiplication and subsequent sum over the feature dimension, of the three input vectors: query $Q$, key $K$ and distance projection $D^K$:

$$
Q = W^Q x_i \quad \text{and} \quad K = W^K x_i
\tag{6}
$$

$$
\text{dot}(Q, K, D^K) = \sum_k^F Q_k \odot K_k \odot D_k^K
\tag{7}
$$

The resulting matrix is passed through a nonlinear activation function and is weighted by a cosine cutoff $\phi$ (see equation 2), ensuring that atoms with a distance larger than $d_{\text{cut}}$ do not interact.

$$
A = \text{SiLU}(\text{dot}(Q, K, D^K)) \cdot \phi(d_{ij})
\tag{8}
$$

Traditionally, the resulting attention matrix $A$ is passed through a softmax activation, however, we replace this step with a SiLU function to preserve the distance cutoff. The softmax scaling factor of $\sqrt{d_k}^{-1}$, which normally rescales small gradients from the softmax function, is left out. Work by Choromanski et al. (2021) suggests that replacing the softmax activation function in Transformers with ReLU-like functions might even improve accuracy, supporting the idea of switching to SiLU in this case.

We place a continuous filter graph convolution (Schütt et al., 2017b) in the attention mechanism's value pathway. This enables the model to not only consider interatomic distances in the attention weights but also incorporate this information into the feature vectors directly. The resulting representation is split into three equally sized vectors $s_{ij}^1, s_{ij}^2, s_{ij}^3 \in \mathbb{R}^F$. The vector $s_{ij}^3$ is scaled by the attention matrix $A$ and aggregated over the value-dimension, leading to an updated list of feature vectors. The linear transformation $O$ is used to combine the attention heads' outputs into a single feature vector $y_i \in \mathbb{R}^{384}$.

$$
s_{ij}^1, s_{ij}^2, s_{ij}^3 = \text{split}(V_j \odot D^V{}_{ij})
$$

$$
y_i = O\left(\sum_j^N A_{ij} \cdot s_{ij}^3\right)
\tag{9}
$$

The attention mechanism's output, therefore, corresponds to the updated scalar feature vectors $y_i$ and scalar filters $s_{ij}^1$ and $s_{ij}^2$, which are used to weight the directional information inside the update layer.

## 2.4 UPDATE LAYER

The update layer (Figure 1b) is used to compute interactions between atoms (attention block) and exchange information between scalar and vector features. The updated scalar features $y_i$ from the attention block are split up into three feature vectors $q_i^1, q_i^2, q_i^3 \in \mathbb{R}^F$. The first feature vector, $q_i^1$, takes the role of a residual around the scaled vector features. The resulting scalar feature update $\Delta x_i$ of this update layer is then defined as

$$
\Delta x_i = q_i^1 + q_i^2 \odot \langle U_1 \vec{v}_i, U_2 \vec{v}_i \rangle
\tag{10}
$$

where $\langle U_1 \vec{v}_i, U_2 \vec{v}_i \rangle$ denotes the scalar product of vector features $\vec{v}_i$, transformed by linear projections $U_1$ and $U_2$.

On the side of the vector features, scalar information is introduced through a multiplication between $q_i^3$ and a linear projection of the vector features $U_3 \vec{v}_i$. The representation is updated with equivariant features using the directional vector between two atoms. The edge-wise directional information is multiplied with scalar filter $s_{ij}^2$ and added to the rescaled vector features $s_{ij}^1 \cdot \vec{v}_j$. The result is aggregated inside each atom, forming $\vec{w}_i$. The final vector feature update $\Delta \vec{v}_i$ for the current update layer is then produced by adding the weighted scalar features to the equivariant features $\vec{w}_i$.

$$\vec{w}_i = \sum_j^N s_{ij}^1 \odot \vec{v}_j + s_{ij}^2 \odot \frac{\vec{r}_i - \vec{r}_j}{\|\vec{r}_i - \vec{r}_j\|} \tag{11}$$

$$\Delta \vec{v}_i = \vec{w}_i + q_i^3 \odot U_3 \vec{v}_i$$

## 2.5 TRAINING

Models are trained using mean squared error loss and the Adam optimizer (Kingma & Ba, 2017) with parameters $\beta_1 = 0.9$, $\beta_2 = 0.999$ and $\epsilon = 10^{-8}$. Linear learning rate warm-up is applied as suggested by Vaswani et al. (2017) by scaling the learning rate with $\xi = \frac{\text{step}}{n_{\text{steps}}}$. After the warm-up period, we systematically decrease the learning rate by scaling with a decay factor upon reaching a plateau in validation loss. The learning rate is decreased down to a minimum of $10^{-7}$. We found that weight decay and dropout do not improve generalization in this context. When training on energies and forces, we apply exponential smoothing to the energy's train and validation loss. New losses are discounted with a factor of $\alpha = 0.05$. See Appendix A for a more comprehensive summary of hyperparameters.

## 3 EXPERIMENTS AND RESULTS

We evaluate the equivariant Transformer on the QM9 (Ramakrishnan et al., 2014), MD17 (Chmiela et al., 2017) and ANI-1 (Smith et al., 2017b) benchmark datasets. QM9 comprises 133,885 small organic molecules with up to nine heavy atoms of type C, O, N, and F. It reports computed geometric, thermodynamic, energetic, and electronic properties for locally optimized geometries. As suggested by the authors, we used a revised version of the dataset, which excludes 3,054 molecules due to failed geometric consistency checks. The remaining molecules were split into a training set with 110,000 and a validation set with 10,000 samples, leaving 10,831 samples for testing.

Table 1 compares the equivariant Transformer's results on QM9 with the invariant architectures SchNet (Schütt et al., 2018), PhysNet (Unke & Meuwly, 2019) and DimeNet++ (Klicpera et al., 2020a), the covariant Cormorant (Anderson et al., 2019) architecture and equivariant EGNN (Satorras et al., 2021), LieTransformer (we compare to their best variant, LieTransformer-T3+SO3 Aug) (Hutchinson et al., 2020) and PaiNN (Schütt et al., 2021). We use specialized output models for two of the QM9 targets, which add certain features directly to the prediction. For the molecular dipole moment $\mu$, both scalar and vector features are used in the final calculation. The output MLP consists of two gated equivariant blocks (Weiler et al., 2018; Schütt et al., 2021) with the same layer sizes as in the otherwise used output network. The updated scalar features $x_i$ and vector features $\vec{v}_i$ are then used to compute $\mu$ as

$$\mu = \left\| \sum_i^N \vec{v}_i + x_i (\vec{r}_i - \vec{r}) \right\| \tag{12}$$

where $\vec{r}$ is the center of mass of the molecule. For the prediction of the electronic spatial extent $\langle R^2 \rangle$, scalar features are transformed using gated equivariant blocks as described above, yielding scalar atomic predictions $x_i$, and multiplied by the squared norm of atomic positions

$$\langle R^2 \rangle = \sum_i^N x_i \|\vec{r}_i\|^2 \tag{13}$$

The MD17 dataset consists of molecular dynamics (MD) trajectories of small organic molecules, including both energies and forces. In order to guarantee conservation of energy, forces are predicted using the negative gradient of the energy with respect to atomic coordinates $\vec{F}_i = -\partial \hat{E}/\partial \vec{r}_i$. To

Table 1: Results on all QM9 targets and comparison to previous work. Scores are reported as mean absolute errors (MAE). LieTF refers to the best performing variant of LieTransformers (Hutchinson et al., 2020), i.e. LieTransformer-T3+SO3 Aug.

| Target | Unit | SchNet | EGNN | PhysNet | LieTF | DimeNet++ | Cormorant | PaiNN | ET |
|---|---|---|---|---|---|---|---|---|---|
| $\mu$ | $D$ | 0.033 | 0.029 | 0.0529 | 0.041 | 0.0297 | 0.038 | 0.012 | **0.011** |
| $\alpha$ | $a_0^3$ | 0.235 | 0.071 | 0.0615 | 0.082 | 0.0435 | 0.085 | **0.045** | 0.059 |
| $\epsilon_{HOMO}$ | $meV$ | 41 | 29 | 32.9 | 33 | 24.6 | 34 | 27.6 | **20.3** |
| $\epsilon_{LUMO}$ | $meV$ | 34 | 25 | 24.7 | 27 | 19.5 | 38 | 20.4 | **17.5** |
| $\Delta\epsilon$ | $meV$ | 63 | 48 | 42.5 | 51 | **32.6** | 61 | 45.7 | 36.1 |
| $\langle R^2 \rangle$ | $a_0^2$ | 0.073 | 0.106 | 0.765 | 0.448 | 0.331 | 0.961 | 0.066 | **0.033** |
| $ZPVE$ | $meV$ | 1.7 | 1.55 | 1.39 | 2.10 | **1.21** | 2.027 | 1.28 | 1.84 |
| $U_0$ | $meV$ | 14 | 11 | 8.15 | 17 | 6.32 | 22 | **5.85** | 6.15 |
| $U$ | $meV$ | 19 | 12 | 8.34 | 16 | 6.28 | 21 | **5.83** | 6.38 |
| $H$ | $meV$ | 14 | 12 | 8.42 | 17 | 6.53 | 21 | **5.98** | 6.16 |
| $G$ | $meV$ | 14 | 12 | 9.4 | 19 | 7.56 | 20 | **7.35** | 7.62 |
| $C_v$ | $\frac{cal}{mol\,K}$ | 0.033 | 0.031 | 0.028 | 0.035 | **0.023** | 0.026 | 0.024 | 0.026 |

Table 2: Results on MD trajectories from the MD17 dataset. Scores are given by the MAE of energy predictions (kcal/mol) and forces (kcal/mol/Å). NequIP does not provide errors on energy, for PaiNN we include the results with lower force error out of training only on forces versus on forces and energy. Benzene corresponds to the dataset originally released in Chmiela et al. (2017), which is sometimes left out from the literature. ET results are averaged over three random splits.

| Molecule | | SchNet | PhysNet | DimeNet | PaiNN | NequIP | ET |
|---|---|---|---|---|---|---|---|
| Aspirin | *energy* | 0.37 | 0.230 | 0.204 | 0.167 | - | **0.123** |
| | *forces* | 1.35 | 0.605 | 0.499 | 0.338 | 0.348 | **0.253** |
| Benzene | *energy* | 0.08 | - | 0.078 | - | - | **0.058** |
| | *forces* | 0.31 | - | **0.187** | - | **0.187** | 0.196 |
| Ethanol | *energy* | 0.08 | 0.059 | 0.064 | 0.064 | - | **0.052** |
| | *forces* | 0.39 | 0.160 | 0.230 | 0.224 | 0.208 | **0.109** |
| Malondialdehyde | *energy* | 0.13 | 0.094 | 0.104 | 0.091 | - | **0.077** |
| | *forces* | 0.66 | 0.319 | 0.383 | 0.319 | 0.337 | **0.169** |
| Naphthalene | *energy* | 0.16 | 0.142 | 0.122 | 0.116 | - | **0.085** |
| | *forces* | 0.58 | 0.310 | 0.215 | 0.077 | 0.097 | **0.061** |
| Salicylic Acid | *energy* | 0.20 | 0.126 | 0.134 | 0.116 | - | **0.093** |
| | *forces* | 0.85 | 0.337 | 0.374 | 0.195 | 0.238 | **0.129** |
| Toluene | *energy* | 0.12 | 0.100 | 0.102 | 0.095 | - | **0.074** |
| | *forces* | 0.57 | 0.191 | 0.216 | 0.094 | 0.101 | **0.067** |
| Uracil | *energy* | 0.14 | 0.108 | 0.115 | 0.106 | - | **0.095** |
| | *forces* | 0.56 | 0.218 | 0.301 | 0.139 | 0.173 | **0.095** |

evaluate the architecture's performance in a limited data setting, the model is trained on only 1000 samples from which 50 are used for validation. The remaining data is used for evaluation and is the basis for comparison with other work. Separate models are trained for each molecule using a combined loss function for energies and forces where the energy loss is multiplied with a factor of 0.2 and the force loss with 0.8. An overview of the results and comparison to the invariant models SchNet (Schütt et al., 2017b), PhysNet (Unke & Meuwly, 2019) and DimeNet (Klicpera et al., 2020b), as well as the equivariant architectures PaiNN (Schütt et al., 2021) and NequIP (Batzner et al., 2021) can be found in Table 2.

To evaluate the architecture's capabilities on a large collection of off-equilibrium conformations, we train and evaluate the equivariant Transformer on the ANI-1 dataset. It contains 22,057,374 configurations of 57,462 small organic molecules with up to 8 heavy atoms and atomic species H, C, N, and O. The off-equilibrium data points are generated via exhaustive normal mode sampling of the energy minimized molecules. The model is fitted on DFT energies from 80% of the dataset,

while 5% are used as validation and the remaining 15% of the data make up the test set. Figure 2 compares the equivariant Transformer's performance to previous methods DTNN (Schütt et al., 2017a), SchNet (Schütt et al., 2017b), MGCN (Lu et al., 2019) and ANI (Smith et al., 2017a).

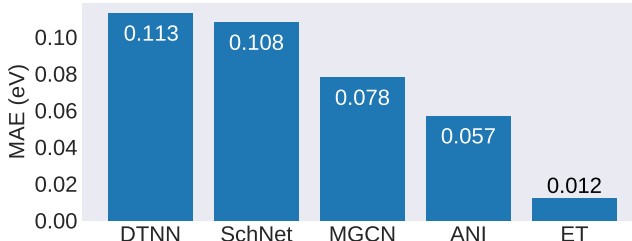

Figure 2: Comparison of testing MAE on the ANI-1 dataset in eV. Results for DTNN, SchNet and MGCN are provided by Lu et al. (2019). The ANI method refers to the ANAKIN-ME (Smith et al., 2017a) model used for constructing the ANI-1 dataset.

### 3.1 ATTENTION WEIGHT ANALYSIS

Neural network predictions are notoriously difficult to interpret due to the complex nature of the learned transformations. To shed light into the black box predictor, we extract and analyze the equivariant Transformer's attention weights. We run inference on the ANI-1, QM9, and MD17 test sets for all molecules and extract each sample's attention matrix from all attention heads in all layers. Attention rollout (Abnar & Zuidema, 2020) under the single head assumption is applied during the extraction, resulting in a single attention matrix per sample. Figure 4 visualizes these attention weights for random QM9 molecules (see Appendix F for further examples).

To analyze patterns in the interaction of different chemical elements, we average the attention weights over each unique combination of interacting atom types (hydrogen, carbon, oxygen, nitrogen, fluorine). This generates two attention scores for each pair of atom types, one from the perspective of atom type $z_1$ attending $z_2$ and vice versa. The attention scores are compared to the probability of this bond occurring in the respective dataset, making sure the network's attention is not simply proportional to the relative frequency of the interaction. Figure 3 presents a summary of these bond probabilities and attention scores for QM9, ANI-1, and the average attention scores for all MD17 models. For further details, see Appendix H.

Since the equivariant Transformer is trained to predict the energy of a certain molecular conformation, we expect it to pay attention to atoms that are displaced from the equilibrium conformation, i.e. the energy-minimized structure, of the molecule. We test this hypothesis by comparing the attention weights of displaced atoms to those of the equilibrium conformation. Using the QM9 test set as the source of equilibrium conformations, we displace single atoms by 0.4Å in a random direction and compare the absolute magnitude of attention weights involving the displaced atom to the remaining attention weights. We find increased attention for displaced carbon and oxygen atoms in all models, however, only the model trained on ANI-1 attends more to displaced hydrogen atoms than to hydrogen in its equilibrium position. Attention for displaced hydrogen atoms even decreases in the model trained on QM9, which suggests that the energy labels in QM9 do not depend strongly on the exact location of hydrogen atoms. For a detailed overview of the results, see Appendix C.

It is interesting to see that the training dataset influences attention. For static structures, like in QM9, attention analysis shows that very little importance is attributed to hydrogens, while core structural atoms like carbons are very important. For datasets which have dynamical data like ANI-1 and MD17, we see that hydrogen attended strongly. This is consistent with the fact that hydrogens are important for hydrogen bond-type interactions and therefore important for dynamics. This suggests that the network is not only learning meaningful chemical representation but also that training on dynamical datasets is important.

### 3.2 ABLATION STUDIES

We quantify the effectiveness of the neighbor embedding layer by the change in accuracy when ablating this architectural component. The neighbor embedding layer is replaced by a regular atom type embedding, which we evaluate by comparing the testing MAE on $U_0$ from QM9. The ablation causes a drop in accuracy of roughly 6% (from 6.24 to 6.60 meV). We also try replacing the neighbor embedding by an additional update layer as the neighbor embedding resembles a graph convolution

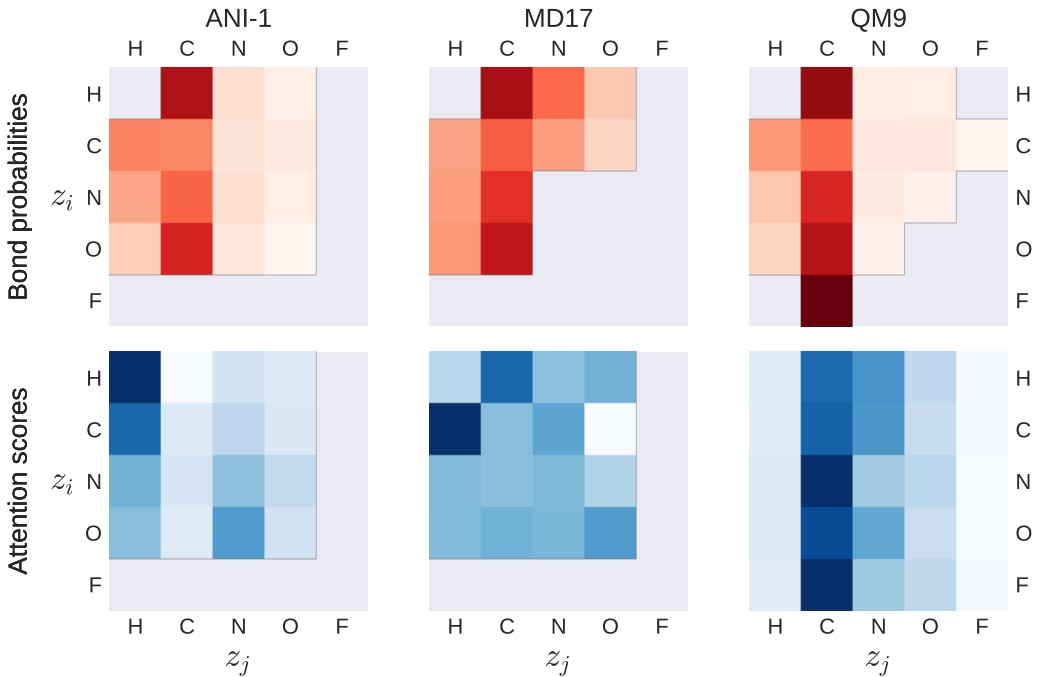

Figure 3: Depiction of bond probabilities and attention scores extracted from the ET model of TorchMD-NET using QM9 (total energy $U_0$), MD17 (average over 8 discussed molecules) and ANI-1 testing data. Attention scores are given as $z_i$ attending $z_j$, bond probabilities follow the same idea, showing the conditional probability of a bond between $z_i$ and $z_j$, given $z_i$. Darker colors correspond to larger values, element pairs without data are grayed out. See Appendix G for an overview of elemental composition in the respective datasets.

operation, which is the equivalent operation to an update layer in graph convolutional networks such as SchNet. Here, the drop in accuracy is even more pronounced with a decrease of around 10% (from 6.24 to 6.85 meV). However, this may also be the result of overfitting as each update layer has about 4.6 times more parameters than a neighbor embedding layer.

The hyperparameter set used for MD17 and ANI-1 results in a model size of 1.34M trainable parameters, which is significantly more than recent similar architectures such as PaiNN (600k) or NequIP (290k). To rule out the possibility that the ET results are simply caused by an increased number of parameters, we train smaller versions of the ET, which are comparable to the size of PaiNN and NequIP. We find that smaller versions of the ET are still competitive and outperform previous state-of-the-art results on MD17. See Appendix D for details on the results, hyperparameters and computational efficiency.

## 3.3 COMPUTATIONAL EFFICIENCY

To assess the computational efficiency of the equivariant Transformer, we measure the inference time of random QM9 batches comprising 50 molecules (including computing pairwise distances) on an NVIDIA V100 GPU (see Table 3). We report times for different sizes of the model, differing in the number of update layers, the feature dimension, and the size of the RBF distance expansion. ET-large uses the QM9 hyperparameter set, while ET-small is constructed using MD17/ANI-1 hyperparameters (see Table 4). The measurements were conducted using just-in-time (JIT) compiled models. The JIT-compiled versions of ET-small and ET-large are 1.5 and 1.8 times respectively more efficient than the raw implementation. We only measure the duration of the forward pass, excluding the backward pass required to predict forces. The force prediction decreases inference speed by approximately 75.9%, resulting in 39.0ms per batch using ET-small and 48.5ms using ET-large.

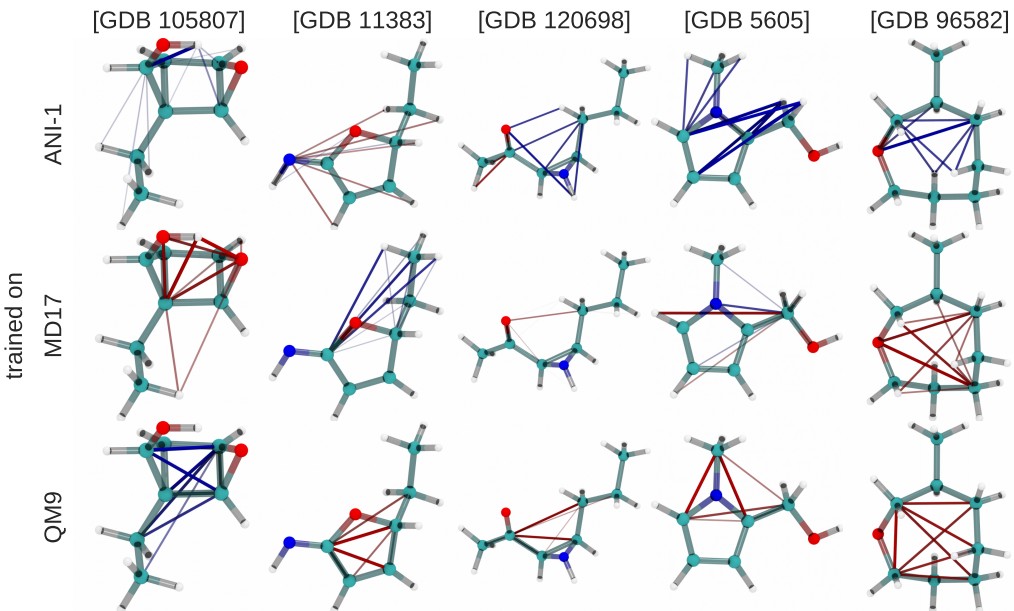

Figure 4: Visualization of five molecules from the QM9 dataset with attention scores corresponding to models trained on ANI-1, MD17 (uracil) and QM9. Blue and red lines represent negative and positive attention scores respectively.

Table 3: Comparison of computational efficiency between PaiNN, DimeNet++ and different sizes of TorchMD-NET ET. The time is measured at inference using random batches of 50 molecules from QM9. Speed of ET models of TorchMD-NET is reported as mean $\pm$ standard deviation over 1000 calls. Values for PaiNN and DimeNet++ are taken from Schütt et al. (2021) so differences in efficiency may to some degree originate from different implementations.

|  | PaiNN | DimeNet++ | ET-small | ET-large |
|---|---|---|---|---|
| time per batch | 13 ms | 45 ms | **9.4ms** $\pm$ 3.4ms | 11.7ms $\pm$ 4.0ms |
| no. parameters | 600k | 1.8M | 1.34M | 6.87M |

## 4 DISCUSSION

In this work, we introduce a novel attention-based architecture for the prediction of quantum mechanical properties, leveraging the use of rotationally equivariant features. We show a high degree of accuracy on the QM9 benchmark dataset, however, the architecture's effectiveness is particularly clear when looking at the prediction of energies and atomic forces in the context of molecular dynamics. We set a new state-of-the-art on all MD17 targets (except force prediction of the molecule Benzene) and demonstrate the architecture's ability to work in a low data regime. As described in previous work (Schütt et al., 2021; Fuchs et al., 2020), the model's vector features and equivariance can be utilized in the prediction of variables beyond scalars. Here, only the dipole moment is predicted in this fashion, however, the architecture is capable of predicting tensorial properties. By extracting and analyzing the model's attention weights, we gain insights into the molecular representation, which is characterized by the nature of the corresponding training data. We show that the model does not pay much attention to the location of hydrogen when trained only on energy-minimized molecules, while a model trained on data including off-equilibrium conformations focuses to a large degree on hydrogen. Furthermore, we validate the learned representation by analyzing attention weights involving atoms displaced from their equilibrium location. We demonstrate that off-equilibrium conformations in the training data play an important role in the accurate prediction of a molecule's energy. This highlights the importance of configurational diversity in the evaluation of neural network potentials.

After the final review of this paper, NequIP's preprint (Batzner et al., 2021) updated the results with better accuracy for MD17. However, it requires using high order spherical harmonics which are likely substantially slower than Transformer models.

## SOFTWARE AND DATA

The equivariant Transformer is implemented in PyTorch (Paszke et al., 2019), using PyTorch Geometric (Fey & Lenssen, 2019) as the underlying framework for geometric deep learning. Training is done using pytorch-lightning (Falcon & The PyTorch Lightning team, 2019), a high-level interface for training PyTorch models. The datasets QM9[1], MD17[2] and ANI-1[3] are publicly available and all source code for training, running and analyzing the models presented in this work is available at github.com/torchmd/torchmd-net.

### ACKNOWLEDGMENTS

GDF thanks the project PID2020-116564GB-I00 funded by MCIN/ AEI /10.13039/501100011033, Unidad de Excelencia María de Maeztu" funded by the MCIN and the AEI (DOI: 10.13039/501100011033) Ref: CEX2018-000792-M and the European Union's Horizon 2020 research and innovation programme under grant agreement No. 823712. Research reported in this publication was supported by the National Institute of General Medical Sciences (NIGMS) of the National Institutes of Health under award number GM140090.

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

## A    ARCHITECTURAL DETAILS AND HYPERPARAMETERS

All models in this work were trained using distributed training across two NVIDIA RTX 2080 Ti GPUs, using the DDP training protocol. This leads to training times of ~16h for QM9, ~10h for MD17 and ~83h for ANI-1.

Table 4: Comparison of various hyperparameters used for QM9, MD17 and ANI-1.

| Parameter | QM9 | MD17 | ANI-1 |
|---|---|---|---|
| initial learning rate | $4 \cdot 10^{-4}$ | $1 \cdot 10^{-3}$ | $7 \cdot 10^{-4}$ |
| lr patience (epochs) | 15 | 30 | 5 |
| lr decay factor | 0.8 | 0.8 | 0.5 |
| lr warmup steps | 10,000 | 1,000 | 10,000 |
| batch size | 128 | 8 | 2048 |
| no. layers | 8 | 6 | 6 |
| no. RBFs | 64 | 32 | 32 |
| feature dimension | 256 | 128 | 128 |
| no. parameters | 6.87M | 1.34M | 1.34M |

While the ET model follows a similar idea as the SE(3)-Transformer introduced by Fuchs et al. (2020), there are significant architectural differences. The SE(3)-Transformer relies heavily on expensive features, such as Clebsch-Gordan coefficients and spherical harmonics while the ET model only requires interatomic distances. Additionally, we split scalar and equivariant features into two pathways, which exchange information inside the update layer while the SE(3)-Transformer computes message passing updates for each type of feature vector (scalar or equivariant). Finally, our modified attention mechanism and update step differ significantly from the SE(3)-Transformer's message passing layer, which, for example, does not handle self-interactions in the attention mechanism and applies only linear transformations to distance features.

## B    IMPORTANCE OF HYDROGEN

While the ET model trained on QM9 mostly attends to carbon atoms, models trained on molecular dynamics trajectories show a strong focus on carbon-hydrogen interactions. This highlights the different nature of datasets containing only energy minimized conformations in contrast to datasets containing MD data. We further support this hypothesis by comparing the reduction in accuracy for models, which are trained without hydrogen, on the datasets QM9 (total energy $U_0$) and MD17 (aspirin). We show that the loss in accuracy when predicting the energy using the model trained on MD17 is one order of magnitude larger than when training only on molecules in the ground state. Furthermore, the accuracy of force predictions drops by another 1.5x compared to MD17 energy predictions when excluding hydrogen. A summary of the results is given in Table 5.

Table 5: Test MAE of the TorchMD-NET ET on QM9 and MD17 (aspirin), trained with and without hydrogen.

| Dataset | | **with** hydrogen | **without** hydrogen | relative change |
|---|---|---|---|---|
| QM9 | | 6.37 meV | 20.83 meV | 227.0% |
| MD17 | *energy* | 5.33 meV | 137.59 meV | 2481.4% |
| | *forces* | 10.67 meV/Å | 433.99 meV/Å | 3966.5% |

## C    ATOM DISPLACEMENT

Figure 5 shows averaged absolute attention scores for molecules where a single atom has been displaced from the equilibrium structure. This reiterates the idea that the model trained only on equilibrium structures (QM9) focuses on carbon and neglects hydrogen. The models trained on off-equilibrium conformations show a higher degree of attention for displaced hydrogen atoms. We

restrict the molecules to only contain hydrogen, carbon, and oxygen in order to compare results between models trained on QM9, MD17 (aspirin), and ANI-1.

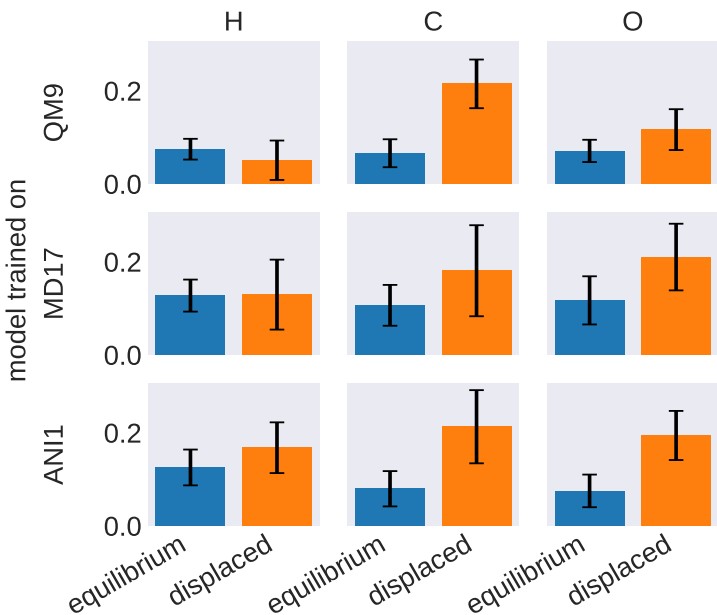

Figure 5: Averaged attention weights extracted from the ET on the QM9 test set (molecules consisting of H, C, and O only) with a displacement of 0.4Å in single atoms. Blue bars show attention towards atoms in equilibrium locations, orange bars correspond to attention weights involving the displaced atom. Attention scores are normalized inside each molecule. The black bars show the attention weights' standard deviation.

## D  EQUIVARIANT TRANSFORMERS WITH REDUCED PARAMETER COUNT

We compare the ET model with a reduced number of parameters, matching those of PaiNN (600k) and NequIP (290k), to state-of-the-art models on the MD17 benchmark. This ensures that our results are not solely a consequence of a larger model size but correspond to an improved architecture. The smaller ET models are still competitive and outperform state-of-the-art on most MD17 targets. Table 6 provides an overview of the adjusted architectural hyperparameters and Table 7 summarizes the results on MD17.

Table 6: Hyperparameter set of the full ET model compared to PaiNN- and NequIP-sized variants. This table only includes hyperparameters that were changed.

| Hyperparameter | full ET | PaiNN-sized ET | NequIP-sized ET |
|---|---|---|---|
| no. layers | 6 | 3 | 3 |
| no. RBFs | 32 | 16 | 16 |
| feature dimension | 128 | 120 | 80 |
| no. parameters | 1.34M | 593k | 273k |

Table 7: Energy (kcal/mol) and force (kcal/mol/Å) MAE of the PaiNN- and NequIP-sized ET models. The "full ET" column is equal to the results in Table 2 and is meant for comparison with the smaller ET variants. Values in bold indicate the best result out of the two models in direct comparison.

| Molecule | | full ET | PaiNN-sized | | NequIP-sized | |
|---|---|---|---|---|---|---|
| | | | ET (594k) | PaiNN (600k) | ET (273k) | NequIP (290k) |
| Aspirin | *energy* | 0.124 | **0.138** | 0.167 | 0.143 | - |
| | *forces* | 0.255 | **0.334** | 0.338 | **0.337** | 0.348 |
| Benzene | *energy* | 0.056 | 0.057 | - | 0.063 | - |
| | *forces* | 0.201 | 0.197 | - | 0.189 | **0.187** |
| Ethanol | *energy* | 0.054 | **0.053** | 0.064 | 0.053 | - |
| | *forces* | 0.116 | **0.112** | 0.224 | **0.123** | 0.208 |
| Malondialdehyde | *energy* | 0.079 | **0.080** | 0.091 | 0.080 | - |
| | *forces* | 0.176 | **0.209** | 0.319 | **0.218** | 0.337 |
| Naphthalene | *energy* | 0.085 | **0.084** | 0.116 | 0.085 | - |
| | *forces* | 0.060 | 0.080 | **0.077** | **0.080** | 0.097 |
| Salicylic Acid | *energy* | 0.094 | **0.095** | 0.116 | 0.097 | - |
| | *forces* | 0.135 | **0.175** | 0.195 | **0.184** | 0.238 |
| Toluene | *energy* | 0.074 | **0.075** | 0.095 | 0.075 | - |
| | *forces* | 0.066 | **0.088** | 0.094 | **0.091** | 0.101 |
| Uracil | *energy* | 0.096 | **0.094** | 0.106 | 0.096 | - |
| | *forces* | 0.094 | **0.122** | 0.139 | **0.128** | 0.173 |

## E   ABLATION OF EQUIVARIANT FEATURES

We perform an ablation of the TorchMD-NET ET's equivariance and compare the performance of the resulting rotationally invariant model to that of the equivariant Transformer. Without equivariance, the MAE in total energy $U_0$ in QM9 increases from 6.24 to 6.64 meV (6%). On aspirin inside the MD17 benchmark, removing the equivariance causes the energy MAE to rise from 5.37 to 13.23 meV (146%), while force errors go up from 11.05 to 30.27 meV/Å (174%). As, without equivariance, the error increases much more drastically on dynamical data, we hypothesize that equivariant features are particularly useful when dealing with non-zero forces.

## F   MOLECULAR REPRESENTATION BY DATASET

Figures 6a, 6b and 6c show a three dimensional visualization of three random molecules from the datasets QM9, MD17 (aspirin) and ANI-1 respectively. We extract the attention weights from the best performing equivariant Transformer on the test sets of the three datasets respectively to make sure that no model has seen a visualized conformation during training. The red and blue lines between atoms depict the 10 largest absolute attention weights where the line width and alpha value represent the absolute attention weight. Red lines show positive attention weights, blue lines correspond to negative attention weights, which occur due to using SiLU activations inside the attention mechanism. While the model trained on QM9 focuses largely on carbon-carbon interactions, it is clear that models trained on MD17 and ANI-1 have a strong focus on hydrogen-carbon and hydrogen-oxygen interactions. This corresponds to the results found in the attention weights averaged over pairwise interactions between elements. The attention weights are normalized inside each molecule.

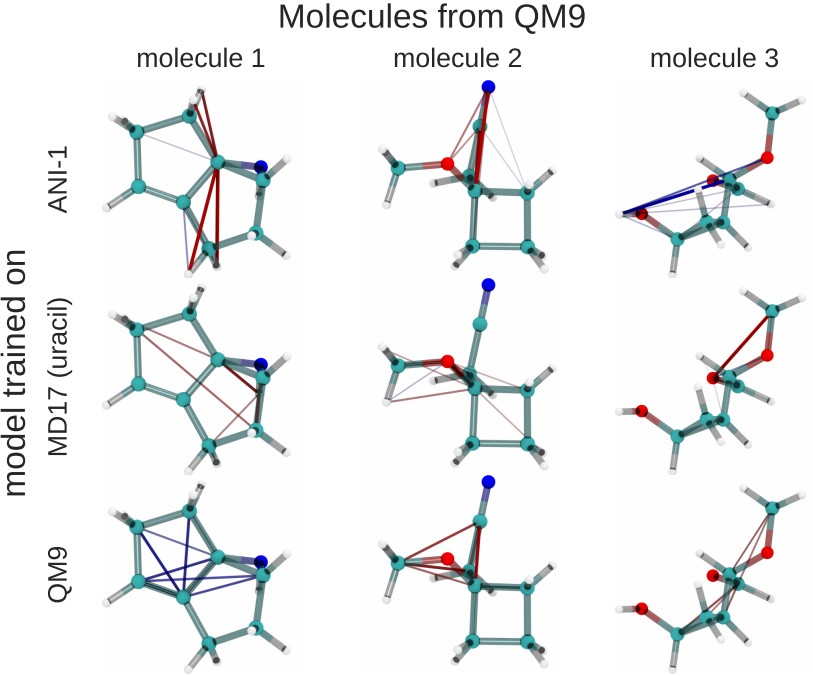

(a) Random molecules from the QM9 test set.

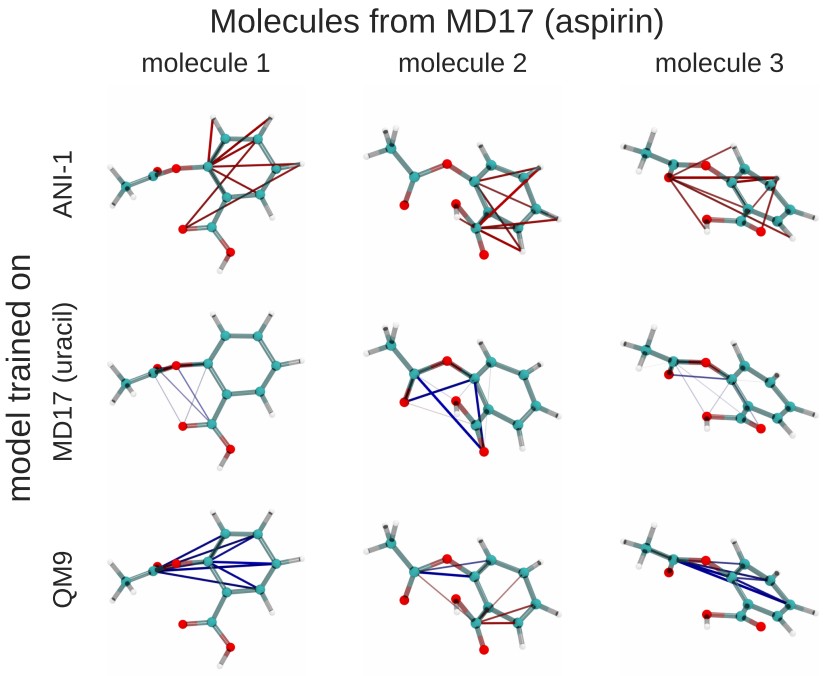

(b) Random conformations from the MD17 test set of aspirin trajectories.

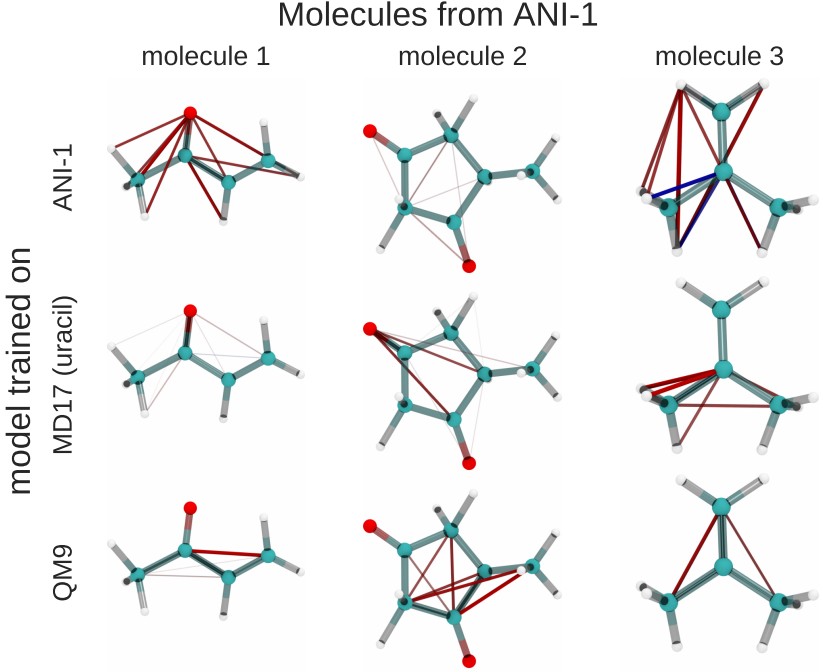

(c) Random molecules from the ANI-1 test set.

Figure 6: Visualization of 10 largest attention weights by absolute value on random molecules from QM9 (a), MD17-aspirin (b) and ANI-1 (c). Each column shows the same molecule, rows correspond to the same ET model trained on QM9, MD17-uracil and ANI-1 respectively.

## G  FREQUENCY OF ELEMENTS

Figure 7 shows the distribution of elements in the datasets QM9, MD17 and ANI-1 where MD17 corresponds to the combination of all target molecules. It aims to assist with the interpretation of attention scores where certain pairs of elements are largely underrepresented in the model's attention. This corresponds predominantly to nitrogen and fluorine as MD17 contains only a single molecule with nitrogen (uracil) and fluorine is only found in QM9.

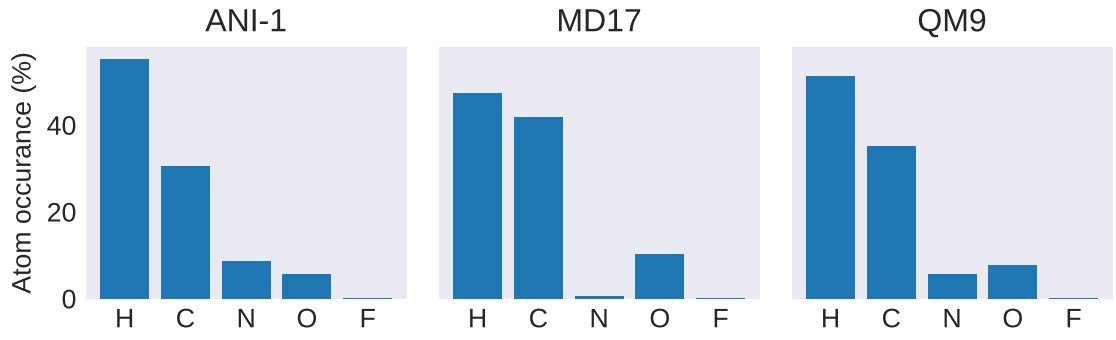

Figure 7: Distribution of elements in the datasets QM9, MD17 (combination of all target molecules) and ANI-1.

# H  MD17 ATTENTION WEIGHTS

Figures 8a and 8b contain the extracted rolled out attention weights for the MD17 target molecules aspirin, benzene, ethanol, malondialdehyde, naphthalene, salicylic acid, toluene and uracil. While the molecules show different attention patterns, a high degree of attention for hydrogen atoms is common for most of the molecules. However, models trained on aspirin, malondialdehyde and uracil additionally exhibit a strong focus on oxygen.

The sum over each row equals one, meaning that the probabilities represent the conditional probability

$$P_{z_i}(\text{bond}_{z_m,z_n}|z_m = z_i) = \frac{N_{\text{bonded}}(z_m, z_n)}{\sum_{z_k \in Z} N_{\text{bonded}}(z_m, z_k)} \tag{14}$$

where $N_{\text{bonded}}(z_i, z_j)$ is the total number of bonds between atom types $z_i$ and $z_j$ found in the collection of molecules.

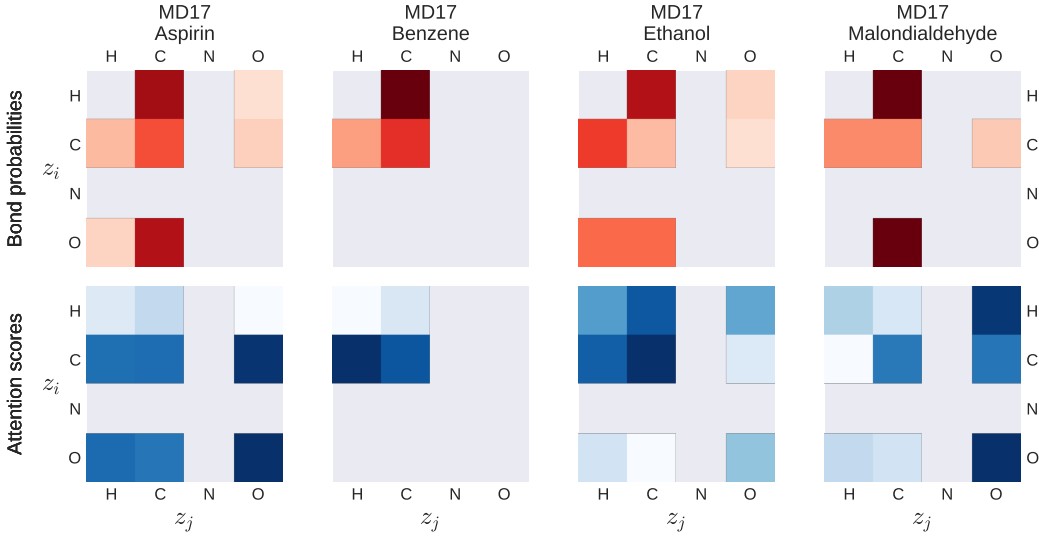

(a) Attention scores and bond probabilities from aspirin, benzene, ethanol and malondialdehyde.

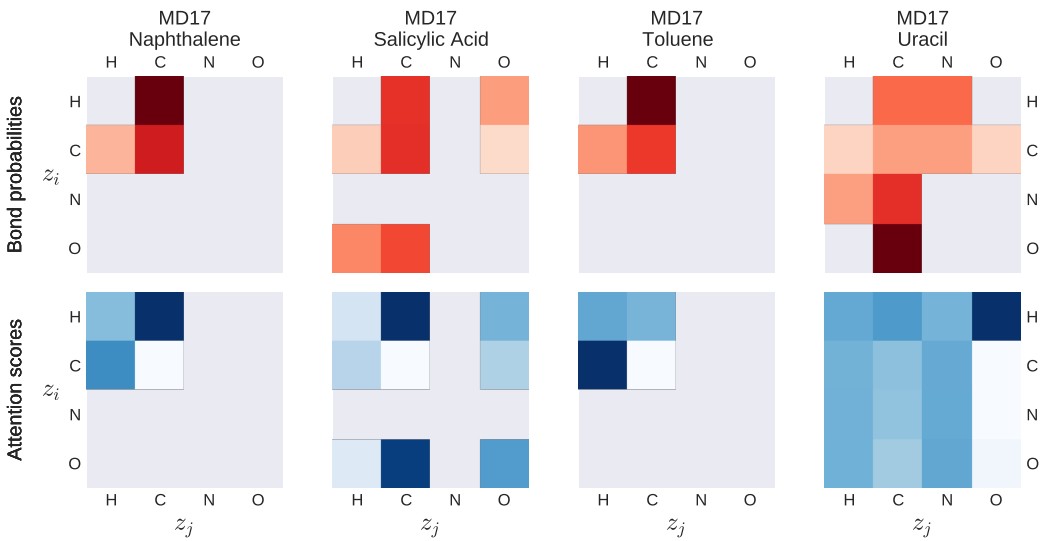

(b) Attention scores and bond probabilities from naphthalene, salicylic acid, toluene and uracil.

Figure 8: Bond probabilities and attention scores extracted from the ET using testing data from all individual molecules in the MD17 dataset. Attention scores are given as $z_i$ attending to $z_j$, bond probabilities follow the same principle, showing the conditional probability of a bond between $z_i$ and $z_j$, given $z_i$. The rightmost subfigure displays the total number of atoms for each type in the data.

## I  PERFORMANCE ON REVISED MD17 TRAJECTORIES

On top of the MD17 dataset, trajectories with higher numerical accuracy were published for some MD17 molecules. This includes aspirin at CCSD and benzene, malondialdehyde, toluene and ethanol at CCSD(T) level of accuracy (Chmiela et al., 2018). Here, we present ET results on these trajectories with the same training protocol as used for the original MD17 dataset (950 training samples, 50 validation samples).

Table 8: ET results on CCSD/CCSD(T) trajectories from Chmiela et al. (2018). Scores are given by the MAE of energy predictions (kcal/mol) and forces (kcal/mol/Å). Results are averaged over two random splits.

|        | Aspirin | Benzene | Ethanol | Malondialdehyde | Toluene |
|--------|---------|---------|---------|-----------------|---------|
| energy | 0.068   | 0.002   | 0.016   | 0.024           | 0.011   |
| forces | 0.268   | 0.008   | 0.103   | 0.168           | 0.062   |

