# OpenReview forum: "Equivariant Transformers for Neural Network based Molecular Potentials"
_ICLR.cc/2022/Conference — ICLR 2022 Spotlight_

### Official Review · Reviewer_ZUVc · 2021-10-26

**Correctness:** 4
**Technical Novelty And Significance:** 3
**Empirical Novelty And Significance:** 3
**Recommendation:** 8
**Confidence:** 4

**Main Review:**

Strengths: The experiment results of this paper looks very promising. In QM9 dataset, the proposed model outperform other models in many targets. Especially for dipole moment and electronic spatial extent, the proposed model is much better. For MD17 dataset, the proposed model outperform all the baselines on all the targets. For ANI-1 dataset, the result is also much better.

Weakness: It will be better to include more discussion about the components of the model, especially the components not in regular transformers. Readers might be curious about
(1) What are the motivations of these components.
(2) Will the performance drop if we replace the components by others? Which components are the most important ones?

Also, there are some other famous baselines such as EGNN [1], DeepPotential [2]. It will be better to include the results in the table.

[1] Satorras, Victor Garcia, Emiel Hoogeboom, and Max Welling. "E (n) equivariant graph neural networks." arXiv preprint arXiv:2102.09844 (2021).
[2] End-to-end symmetry preserving inter-atomic potential energy model for finite and extended systems
L Zhang, J Han, H Wang, WA Saidi, R Car - arXiv preprint arXiv:1805.09003, 2018

**Summary Of The Paper:**

This paper proposed equivariant transformers --- a neural network based algorithm to predict properties of molecules. The architecture is built upon the traditional transformer architecture, combined with some modifications specific to molecular property prediction tasks, such as exponential normal radial basis functions, SiLU activation function and design of update layers. It is shown that the proposed model have good performance on QM9, MD17 and ANI-1 dataset. Ablation studies analyze the attention weight and give some insight about how the model works.

**Summary Of The Review:**

Overall, this paper is a good paper with strong experiment results on various datasets. The components of the model look very interesting. Possible improvement include more discussion on model components and comparison with more baseline algorithms.

---

> ### Author Response · Authors · 2021-11-18
> **Review response**
>
> Thank you for the review.
>
> We added the following to the manuscript:
> “We perform an ablation of the ET’s equivariance and compare the performance of the resulting rotationally invariant model to that of the equivariant Transformer. Without equivariance, the MAE in total energy $U_0$ in QM9 increases from 6.24 to 6.64 meV (6%). On aspirin inside the MD17 benchmark, removing the equivariance causes the energy MAE to rise from 5.37 to 13.23 meV (146%), while force errors go up from 11.05 to 30.27 meV/Å (174%). As, without equivariance, the error increases much more drastically on dynamical data, we hypothesize that equivariant features are particularly useful when dealing with non-zero forces.”
>
> EGNN provides results on QM9 which we have added to the table. While DeepPotential provides results on MD17, it was trained on 50,000 samples. For the benchmark results we report, the number of training samples was restricted to only 1,000.

---

### Official Review · Reviewer_AQ9R · 2021-10-28

**Correctness:** 3
**Technical Novelty And Significance:** 2
**Empirical Novelty And Significance:** 2
**Recommendation:** 6
**Confidence:** 5

**Main Review:**

The paper is well structured and the proposed method is described clearly. The use of the modified attention mechanism could be better motivated. Often, attention is used to capture non-local interactions for which convolutions are not suitable (e.g. Unke et al 2021). Here, the attention is forced to be local by a cutoff function and more reminiscent of factorized tensor layers.

ET improves on the QM9 dataset for 5 of 12 chemical properties. It does not improve however on the energies-related properties (U0, U, G, H), while improvng on energies on the MD17 dataset. This might indicate overfitting. ET uses more than 10x the number of parameters as NequIP or PaiNN. Therefore, it does not become clear whether the improved performance stems from the ET architecture or the size of the model. This should be demonstrated in the ablation studies with reduced number of layers, feature dimensions and/or RBFs.

The analysis of attention scores is interesting, however, it does not become clear from the paper how to interpret the results, e.g. in Figs 3 and 4. In the current form, the shown results are not too useful to "shed light into the black box predictor". The analysis of attention to displaced hydrogen is aligned with what one would expect from equilibrium vs out-of-equilibrium data, but could probably also have been achieved by a sensitivity analysis. The paper could be improved by extracting patterns from the attention scores that, ideally, go beyond pairwise interactions.

The demonstrated computational efficiency is impressive, in particular given the model size. Is there an explanation why the computation time is comparable to the much smaller model except implementation issues? Do the benchmarks include the computation of interatomic distances with scalable neighbor lists, or is the implementation considering all pairwise interactions? A fast evaluation is particularly important for molecular dynamics simulations. Therefore, having a comparison for MD17 molecules with PaiNN and and NequIP would be very relevant.

Further comments:
- p. 1: Pfau et al and Hermann et al have proposed neural network approaches to VMC, not coupled cluster
- p. 1: the original publication of Behler-Parinello should be cited (PRL, 2007) when referring to first work in this field
- p. 2: when using gated equivariant blocks as in PaiNN, Weiler et al 2018 should also be cited who introduced equivariant gated non-linearities

**Summary Of The Paper:**

The paper describes an equivariant neural network (ET) with attention mechanisms that is applied to the prediction of molecular properties. The ET performs competitive with previous approaches on commonly used benchmark data.

**Summary Of The Review:**

The presented approach shows iterative improvements over previous work. Unfortunately, at the current stage it is not clear whether those are due to the ET architecture or the increased model size. The analysis of attention scores is promising, but still lacks the necessary interpretation.

Update:
Based on the response of the authors, I have raised my score.

---

> ### Author Response · Authors · 2021-11-18
> **Review response**
>
> Thank you for the review.
>
> We have added these clarifications in the manuscript:
> “We compare the ET with a reduced number of parameters, matching those of PaiNN (600k) and NequIP (290k), to state-of-the-art models on the MD17 benchmark. This ensures that our results are not solely a consequence of a larger model size but correspond to an improved architecture. The smaller ET models are still competitive and outperform state-of-the-art on most MD17 targets.”
> We also added a table detailing the modified hyperparameter set (number of layers, features dimension, number of RBFs) used to construct the smaller models.
>
> We also performed an ablation study removing the equivariant part:
> “We perform an ablation of the ET’s equivariance and compare the performance of the resulting rotationally invariant model to that of the equivariant Transformer. Without equivariance, the MAE in total energy $U_0$ in QM9 increases from 6.24 to 6.64 meV (6%). On aspirin inside the MD17 benchmark, removing the equivariance causes the energy MAE to rise from 5.37 to 13.23 meV (146%), while force errors go up from 11.05 to 30.27 meV/Å (174%). As, without equivariance, the error increases much more drastically on dynamical data, we hypothesize that equivariant features are particularly useful when dealing with non-zero forces.”
>
> We added a paragraph to further explain key findings of the attention weight analysis. Attention analysis is intrinsically pair-wise so we cannot easily extend it. Compared to a sensitivity analysis, the attention weight analysis additionally provides insights into the latent molecular representation of models trained on equilibrium vs. out-of-equilibrium data.
>
> The computation time is comparable to the much smaller model because the GPU is better utilized on a larger number of parameters during inference.
> The benchmarks do include all computations to perform one step including the computations of interatomic distances.
> NequIP does not provide to the best of our knowledge a benchmark while PaiNN does for QM9. The speed benchmark would not change if we take molecules from MD17 or QM9.
>
> We corrected these errors in the manuscript and added the two missing citations.

---

### Official Review · Reviewer_RnuV · 2021-11-02

**Correctness:** 3
**Technical Novelty And Significance:** 3
**Empirical Novelty And Significance:** 3
**Recommendation:** 8
**Confidence:** 4

**Main Review:**

# Strengths

1. The idea to put attention at the very center of an ML force field is novel.
2.  The performance of the method is state-of-the-art.

# Weaknesses

1. The expose of the attention mechanism and update layer lacks detail.
2. The value of inspecting attention weights is not clear to me.

**Summary Of The Paper:**

The authors introduce a novel architecture for ML force fields, the Equivariant transformer (ET). It is based on the Transformer approach and can be used to predict energies (and forces) and other molecular properties (e.g., QM targets). The performance on standard benchmarks such as QM9 and MD17 is impressive. The authors inspect the attention weights.

**Summary Of The Review:**

1. The introduction is well-written and relates the work to previous works.
2. Sections 2.1 to 2.4 are not detailed enough. Considering that these Sections is at the very center of this work, they deserve more details:
   1) is the function $a_n$ defined anywhere?
   2) where are the matrices (?) Q and K coming from?
   3) how is this attention matrix A computed?
   4) how did the authors arrive at the update rules in equations (8) and below?
3. Specification of the training details is welcome.
4. Did you consider using the updated version of MD17, rMD17, for your experiments? Why did you decide against using the updated dataset?
5. There is a typo in Table 3 (malonaldehyde).
6. In the abstract, the authors claim to have gained "valuable insights into the black box predictor". In the manuscript, the value of their insights is not properly laid out. What did the authors learn about the model that would help them improve it, for instance?

# Update

The authors updated the manuscript and addressed my questions. For this reason, I am willing to increase my score.

---

> ### Author Response · Authors · 2021-11-18
> **Review response**
>
> Thank you for the review.
>
> 2. We cleared up sections 2.1 and 2.4, detailing how every term is computed and what previously unclear symbols stand for (see the updated manuscript):
>     1. we explicitly defined $a_n$, which is the embedding function in the neighbor embedding
>     2. $Q$ and $K$ are the queries and keys respectively, which are computed via multiplication of feature vectors with trainable weight matrices $W^Q$ and $W^K$
>     3. the attention matrix is computed as $ A = \mathrm{SiLU}(\mathrm{dot}(Q, K, D^K)) \cdot \phi(d_{ij}) $
>     4. the equations describe the information exchange between scalar and vector features where the dot product between linear transformations of the vector features scale scalar features (similar to dot-product attention) and scalar features scale vector features (similar to a convolution operation)
> 3. We have added information on the training procedure, hardware and time. All details are  there and hyperparameters in supplementary, while the full training code will be available upon publication.
> 4. We chose to train on MD17 in order to have results comparable to previous methods.
> 5. Malonaldehyde and Malondialdehyde refer to the same molecule. We chose the latter as it seems to be the commonly accepted naming in previous works in this field.
> 6. We have clarified now what is the physical interpretation of the attention model:
> “It is interesting to see that the training dataset influences attention. For static structures, like in QM9, attention analysis shows that very little importance is attributed to hydrogens, while core structural atoms like carbons are very important. For datasets which have dynamical data like ANI-1 and MD17, we see that hydrogen is strongly attended. This is consistent with the fact that hydrogens are important for hydrogen bond-type interactions and therefore important for dynamics. It suggests that the network is not only learning meaningful chemical representation but also that training on dynamical datasets is important.”

---

### Official Review · Reviewer_9RJX · 2021-11-02

**Correctness:** 4
**Technical Novelty And Significance:** 3
**Empirical Novelty And Significance:** Not applicable
**Recommendation:** 6
**Confidence:** 3

**Main Review:**

Strengths
1. The presented transformer model is novel and is applicable to atomic graphs because it is equivariant to rotations and includes edge features in the attention computation. These edge features include an envelope term that prevents discontinuities in the loss landscape. This model is also well motivated by the underlying physics of the problem.
2. The model obtains excellent results on a variety of datasets showing its generality.
3. The paper includes detailed analysis of the learned attention weights.
4. The presented model has good runtime performance compared to models like Dimenet++ that require higher order interactions, while achieving good performance. In practical applications like  catalyst discovery, these models are often used to search through millions of examples making efficient inference important.

Weakness
1. Other equivariant transformers have recently been proposed in the literature (for e.g. SE(3)-Transformers: https://arxiv.org/abs/2006.10503). Since these models are comparable to the method proposed in this paper, it would be good to compare these models. Also, clarifying the differences between these models would make the contributions of this paper clearer.
2. The analysis of attention seems difficult to follow for somebody who is unfamiliar with chemistry terminology. Since ICLR is a general DL conference, I would urge the authors to edit the text to make it easier to follow.


**Summary Of The Paper:**

The paper presents an equivariant transformer model for predicting quantum mechanical properties from an atomic graph. The model obtains SOTA or near-SOTA results on three popular datasets while maintaining good computational efficiency. The primary novelty in their method is a new way to compute the attention score using edge features. The paper also presents a detailed analysis of the attention weights that give insights into what the model is attending over. This is interesting from a chemistry perspective.


**Summary Of The Review:**

The paper presents a new rotation equivariant transformer model for atomic prediction. The resulting method obtains good results on 3 popular datasets, while being computationally efficient at inference time. The paper also provides a nice analysis of the attention weights that shed light on the inner workings of their model. While I would recommend the authors to clarify this analysis for the final version, I am leaning towards accepting the paper in its current form.

---

> ### Author Response · Authors · 2021-11-18
> **Review response**
>
> Thank you for the review.
>
> 1. We have added this clarifying text in the manuscript:
> “While the ET follows a similar idea as the SE(3)-Transformer introduced by Fuchs et al. (2020), there are significant architectural differences. The SE(3)-Transformer relies heavily on expensive features, such as Clebsch-Gordan coefficients and spherical harmonics while the ET only requires interatomic distances. Additionally, we split scalar and equivariant features into two pathways, which exchange information inside the update layer while the SE(3)-Transformer computes message passing updates for each type of feature vector (scalar or equivariant). Finally, our modified attention mechanism and update step differ significantly from the SE(3)-Transformer’s message passing layer, which, for example, does not handle self-interactions in the attention mechanism and applies only linear transformations to distance features.”
>
> 2. We simplified and clarified the chemical terminology in the attention weight analysis by defining atom types and equilibrium structure. This part is important to validate that the model is learning meanful chemical representations.

---

### Decision · Program_Chairs · 2022-01-20

**Decision:**

Accept (Spotlight)

**Comment:**

The paper proposes a rotationally equivariant transformer architecture for predicting molecular properties. The proposed architecture demonstrates good computational efficiency and good results on three benchmarks.

All four reviewers recommend acceptance (two weak, two strong), citing the novelty of the architecture, the good computational efficiency of the model and the good empirical results as the main strengths of the paper. The reviewers expressed minor criticisms and recommendations for improvement, some of which were addressed by the authors during the reviewing process, which led to an increase in scores.

Overall, this is a nice contribution of machine learning to science, and I'm happy to recommend acceptance to ICLR.